# Topical and Intralesional Immunotherapy for Melanoma In Situ: A Review

**DOI:** 10.3390/cancers15184468

**Published:** 2023-09-08

**Authors:** Sandra Martínez-Fernández, Beatriz González-Sixto, Martina Espasandín-Arias, Diego Soto-García, Ángeles Flórez

**Affiliations:** 1Department of Dermatology, Pontevedra University Hospital, 36001 Pontevedra, Spain; beatriz.gonzalez.sixto@sergas.es (B.G.-S.); martina.espasandin.arias@sergas.es (M.E.-A.); diego.soto.garcia@sergas.es (D.S.-G.); angeles.florez.menendez@sergas.es (Á.F.); 2DIPO Research Group, Galicia Sur Health Research Institute (IIS Galicia Sur), SERGAS-UVIGO, 36213 Pontevedra, Spain

**Keywords:** intralesional, immunotherapy, lentigo maligna, melanoma in situ, review, topical

## Abstract

**Simple Summary:**

The incidence of melanoma in situ has increased faster than invasive melanoma over the last decades. Correctly managing these lesions is crucial. The gold standard of treatment for MIS, including lentigo maligna (LM), is complete surgical excision with clear margins (>0.5–1 cm). However, surgery is not always possible, as MIS often affects elderly patients with comorbidities and contraindications for surgical procedures or involves large lesions in functionally sensitive areas. Alternative non-surgical treatments are needed for these cases, which include radiotherapy, cryosurgery, immunotherapy, laser therapy, and other topical medications. This study aims to review the published literature on the applications of immunotherapy in MIS, either in monotherapy or in combination with other therapeutic alternatives.

**Abstract:**

The incidence of in situ melanoma (MIS) has increased over the last decades. The mainstay of treatment for MIS, including lentigo maligna (LM), is complete surgical excision with clear margins (0.5 to 1.0 cm). Nevertheless, MIS lesions often affect elderly patients with comorbidities and involve large lesions in cosmetically sensitive areas, which means surgery is not always appropriate. Non-surgical treatments have a role in these cases, and include radiotherapy, cryosurgery, immunotherapy, laser therapy, and other topical medications. This study aims to review the applications of immunotherapy in MIS, either in monotherapy or in combination with other therapeutic alternatives. The main forms of immunotherapy used are imiquimod and, to a lesser extent, intralesional interferon-α (IL-INF-α) and ingenol mebutate (IM). IL-INF-α and IM have not been studied as extensively as imiquimod, whose results in real-life practice are encouraging. The clearance and recurrence rates reported in MIS treated with imiquimod as monotherapy, or as an adjuvant after surgery with affected or narrow margins, make imiquimod a reliable therapeutic alternative in selected cases. Also, its use as a neoadjuvant therapy before surgery was shown to reduce the final surgical defect size required to confirm negative histologic margins. In conclusion, local immunotherapy is frequently used in clinical practice and experience confirms it to be an excellent option for certain patients.

## 1. Introduction

Cutaneous melanoma (CM) is potentially the most dangerous form of skin tumour and causes 90% of skin cancer mortality [1]. The incidence of CM has risen steadily over the last 4 decades in most Western countries with fair-skinned populations, such as the United States, Europe, or Australia [2,3,4,5,6,7,8,9,10]. The incidence of melanoma in situ (MIS) has also grown through the years, but the rate of growth is higher than that of invasive cases [4,5,6,7,8,9,10]. Olsen et al. analysed long-term incidence trends (1982–2018) for in situ and invasive CM in three populations with high, medium, and low CM ratio: Queensland (Australia), US White, and Scotland [4]. They observed that MIS increased by 7.1–8.4% per year depending on the population, but invasive CM increased slowly, between 1.4 and 3% per year. Moreover, some studies have reported that the incidence of MIS and thin CM has been increasing faster than that of thick CM [5,6]. One explanation for these trends is that improvements in secondary prevention in recent years have shifted the diagnosis of melanoma to the earlier stages, but it could also be an “epidemic of diagnosis” owing to the increased awareness of CM, which leads to overdiagnosis [11,12].

MIS is defined by the presence of neoplastic melanocytes limited to the epidermis (with lentiginous or pagetoid growth patterns) during a non-invasive radial growth phase [13]. It is considered that MIS is a biologic precursor of invasive CM [13]. Therefore, the detection and treatment of MIS would reduce the incidence of invasive tumours and melanoma mortality. Lentigo maligna (LM) is an in situ subtype of CM. Clinically, it appears as a slowly enlarging brown-to-black and sometimes amelanotic asymmetric macule, on chronic sun-exposed skin, and mostly originates on the head and neck of elderly individuals [14]. LM is the most prevalent in situ variant, accounting for 79–83% of all MIS [15]. Approximately 5% of LM develop into invasive melanoma, named LM melanoma (LMM) [16], which represents 2.7–14% of all CM [1]. The major challenges for LM treatment are its main location on the face, with poorly defined limits, and its high risk of local recurrence.

Tzellos et al. performed a Cochrane systematic review on the treatment of MIS, including LM, and concluded that there is a lack of high-quality evidence for both surgical and non-surgical treatments [15]. Different guidelines for melanoma treatment establish that, as in the case of invasive CM, the mainstay of treatment for MIS including lentigo maligna (LM) is complete surgical excision with clear margins [17,18,19,20]. Although no randomised trials have assessed surgical margins for MIS [15], guidelines recommend a wide excision (WE) with 0.5 to 1.0 cm margins [17,18,19,20]. In the LM-subtype, which has a higher propensity for subclinical peripheral tumour extension, margins > 0.5 cm are required to achieve histologically negative margins and, where possible, surgery with microscopic margin control (staged excisions with paraffin-embedded permanent sections—SE or Mohs micrographic surgery—MMS) should be considered [17,18,19,20].

As previously noted, MIS lesions often affect elderly patients with comorbidities (who may have contraindications for surgical procedures) and involve large lesions in cosmetically or functionally sensitive areas (especially in LM). Thus, surgery is not always reasonable or appropriate [17,18,19,21]. Non-surgical treatments have roles in the primary treatment of MIS in patients who are not candidates for surgery (unresectable or inoperable due to high surgical risk: ischemic heart disease, severe heart failure, end-stage renal disease, etc.) or that decline to undergo surgery, but also as an adjuvant to excision when it is incomplete or with narrow margins, or even as a neoadjuvant before surgery [15,21]. They include radiotherapy, cryosurgery, immunotherapy, laser therapy, and other topical medications (5-flurouracil, azelaic acid, or retinoic acid derivates). All these alternative therapies need continuous long-term follow-up to monitor a potential recurrence or invasive component, and they are mainly used in the LM-subtype.

This study aims to review the applications of topical and intralesional immunotherapy in MIS, either in monotherapy or in combination with other therapeutic alternatives.

## 2. Material and Methods

We undertook an exhaustive narrative review of the published literature to revise the uses of topical and intralesional immunotherapy in MIS. The inclusion criteria were defined as follows: (1) articles on the use of immunotherapy in patients with MIS as monotherapy; (2) articles combining immunotherapy with other therapeutic options for MIS; and (3) any study design was accepted. The exclusion criteria included any published article in a language other than English or Spanish.

Several literature databases were analysed in May 2023 (Pubmed, Dialnet, Advanced Google, Trip). Different search strategies were applied, in which both controlled and free language with key words such as melanoma in situ, immunotherapy, imiquimod, ingenol mebutate, intralesional interferon, or immunocryosurgery were used.

The articles were selected in three stages, which were carried out by two reviewers. Initially, the citations retrieved by the search strategies were screened by reading the title, and those that did not meet the inclusion criteria were excluded. In the second stage, the remaining citations were then screened using the abstracts. The selection process was then completed with a detailed reading of each article. Finally, a secondary search of articles included was performed through reading the reference list of the retrieved articles.

## 3. Immunotherapy in Melanoma In Situ

The immune system has been long recognised as a key element involved in the pathogenesis, progression, and persistence of skin cancer. Immunotherapy, which includes various strategies to stimulate and activate the tumour immune response, currently represents a promising option for skin cancer management, including the treatment of MIS [22]. Local immunotherapy is an exciting therapeutic approach that may achieve complete responses without inducing systemic toxicity [22]. It can be used in monotherapy or in combination with other therapies and can be withdrawn and reintroduced as needed without a loss of efficacy. The most widely used immunotherapy in MIS, especially the LM-subtype, is imiquimod and, to a lesser extent, intralesional interferon-α (IL-INF-α) and ingenol mebutate (IM).

### 3.1. Imiquimod

Imiquimod is approved by the European Medicines Agency and US Food and Drug Administration for the treatment of genital warts, actinic keratosis, and superficial basal cell carcinoma [22]. Although it is an off-label therapy, its use for MIS, especially LM, is widely described in the published literature (Table 1 and Table 2). This drug is commercially available as both 3.75% and 5% cream, but the concentration used in the treatment of MIS is 5%.

Imiquimod is a nucleoside analogue of the imidazoquinoline family, with antiviral and antitumoural activity, which acts as a topical immune-response modifier, stimulating innate cutaneous immunity and the cytotoxic arm of the adaptive immune response [23]. The main responsive cell type of its biological effects is the dendritic cell. The said effects are explained by at least three main molecular pathways. The primary action of imiquimod is mediated through agonistic activity towards toll-like receptor (TLR)-7/8 and consequent activation of nuclear factor-kappa B (NF-kB) cascade, leading to a high T-helper (Th1)-weighted antitumoural cellular immune response. The second activity consists of the interference with adenosine receptor signalling pathways, resulting in NF-kB responses independent of TLR signalling. Finally, imiquimod induces apoptosis of tumour cells at higher but therapeutically relevant concentrations [23].

Although it may be well-tolerated, imiquimod has frequent local adverse effects that include erythema, erosions, ulcerations, vesicles, crust, pruritus, pain, and post-inflammatory pigmentation. When applied to the periocular region, it can also cause ocular stinging, swelling, redness, conjunctivitis, and even ectropion, keratitis, or corneal oedema [24,25]. This drug may also cause side effects at non-application sites such as flu-like systemic syndrome or distant inflammatory mucosal reactions [26]. Some long-term adverse effects have been described, such as vitiligo-like depigmentation, associated with the mechanism of the action of imiquimod [27], and lymphoedema, caused by severe inflammation and dermal fibrosis [28].

#### 3.1.1. Imiquimod as Primary Therapy

To our knowledge, the bulk of the published literature regarding imiquimod for MIS or LM are case reports, case series, or prospective cohort studies, and there are only a few non-controlled open-label studies and two randomised clinical trials [29,30] (Table 1). There is no high-quality scientific evidence supporting the use of imiquimod as primary therapy in non-selected cases of MIS [15,18]. However, the published literature supports topical imiquimod as a potential alternative to surgery in selected cases which are not eligible for surgery or radiotherapy [18,19,20]. Although most of the large series and case reports are patients with LM, especially in the facial region, it is also used for all types of MIS, especially in difficult locations where surgery is complex. This is the case of acrolentiginous MIS [31] or MIS in genital regions, the penis and vulva, with or without involvement of the urethra [32,33,34,35,36].

Clearance rates with imiquimod vary widely depending on the study, which is probably due to the variability in the treatment regimen, the duration of the therapy, and the assessment of outcomes (Table 1). Two systematic reviews that analysed published data on the use of imiquimod to treat LM reported similar clinical and histological clearance rates of approximately 78.3% and 76.2–77%, respectively [37,38]. Recent studies support these results. Tio et al., in a prospective cohort of 57 LM, observed complete clinical clearance in 84.2% of patients, and histological clearance in 86% based on targeted biopsies [39]. Flores et al. found a similar histological clearance rate of 83% by staged excision after imiquimod treatment [40]. Two prospective studies, with 34 [41] and 114 patients [42] with LM, described higher rates of clearance with 97% of clinical response, while some retrospective series of 24 [43], 56 [44], and 71 [45], showed a clinical clearance range from 72% to 87%.

These findings contrast with 46% clinical and 37% pathological complete response rates, respectively, achieved in a non-controlled open-label phase II study of 28 LM [46]. In this study, pathological outcomes were classified into four groups: complete pathological response (defined as the absence of LM), partial pathological response (defined as the presence of atypical melanocytes with abnormal distribution and number, but insufficient features to make a diagnosis of LM), no response (presence of LM), and progressive disease (invasive melanoma). If lesions with partial and complete pathological response were analysed together, the efficacy would be similar to the aforementioned studies (74%). Moreover, the excision of the lesions was performed immediately after imiquimod treatment, which is too short to allow a prolonged immunological reaction to destroy additional neoplastic cells, as highlighted by Maher and Guitera [47].

It is worth pointing out that residual clinical hyperpigmentation after imiquimod treatment can be present in correctly treated lesions with histological clearance, which translates into the presence of melanophages and melanin in the dermis [46,48,49,50,51,52,53,54].

The most effective regimen remains undetermined. However, two systematic reviews recommended an intensive protocol of at least 60 applications of imiquimod, with a frequency of 5–7 days per week, which showed the greatest odds of complete clinical and histological clearance [37,38]. In line with this, the regimen that has shown good outcomes in different studies is the application of imiquimod with 1–2 cm margins for 5–7 days per week over 12 weeks, with the objective being to achieve visible inflammation for at least 10 weeks [39,45,50,55]. A clinically apparent inflammatory response during treatment of LMs has been significantly associated with higher clearance rates [43,52,54,56]. In accordance with this, Lallas et al. described 13-fold higher odds of complete clinical response when a robust inflammatory reaction was detected [44]. 

Different strategies have been described to enhance the inflammatory response when necessary, including increasing the frequency of application (>5 days/week, twice daily) or the duration of treatment (>12 weeks), using occlusion with a bandage, the additional application of a topical retinoid, or combining with cryotherapy [55]. On the other hand, if the inflammation is very severe, the application can be reduced to 2–3 days per week [39,50,55].

The long-term recurrence rate is as important as the initial response to imiquimod therapy. Two systematic reviews reported recurrence rates of 2.3% and 2.2% after a mean follow-up of 34.2 [37] and 18.6 [38] months. This is a very low recurrence rate compared to 24.5% described by Read et al. in another systematic review [57]. A recent report by Seyed Jafari et al., with the largest cohort of LM primarily treated with imiquimod to our knowledge (n = 104), reported a long-term follow-up with a median of 8 years and found recurrence rates of 12.6%, 23.5%, and 25.7% at 3, 5, and 10 years, respectively [42]. Chambers et al., in another study (n = 71), with a median follow-up of 5.1 years, reported a recurrence rate of 10.1%, developing at mean 2.9 years [45]. 

The prognosis markers of recurrence after imiquimod treatment for LM were studied by different authors. Gautschi et al. reported that the risk of local recurrence was significantly associated with three histological features in the baseline biopsy specimen: the number of total melanocytes, basal and suprabasal melanocytes, and pagetoid spreading melanocytes [58]. Other authors found a significant association between local recurrence and a history of failed excision, <60 applications of imiquimod, <5 applications per week, and partial clinical clearance [45].

The inability of imiquimod to treat follicular extension or a potential invasive component due to its reduced penetration is a risk factor for progression to LM [19]. In a systematic review of LM treated primarily with imiquimod, the progression to LMM was reported in 1.8% of cases, at an average of 3.9 months during follow-up [38]. Similar rates ranging from 1.3% to 4.5% were reported [29,39,42,48,49,54,56,59], although other studies showed a higher risk of progression (8–9%) [51,60].

Owing to the possibility of clinical clearance without a complete histological response and the risk of recurrence, MIS lesions treated with imiquimod must be biopsied after treatment and followed up for years. It is not recommended to evaluate the treatment failure until 3–6 months after concluding treatment [50,54,55]. It may be possible that the treatment has not completely taken effect before that time and an early biopsy could show persistent disease that may eventually clear up. The mean time to relapse in most studies is greater than 2 years [37,38,39,41,42,45,48,58,60]. Thus, patient follow-up must be prolonged. We agree with some authors who recommend follow-up every 6 months using dermoscopy [53]. Recently, it has been demonstrated that reflectance confocal microscopy (RCM) is a non-invasive tool useful not only for diagnostic LM but also for monitoring the response to imiquimod therapy and detecting relapses [30,41,59,61]. If possible, we can use it as an alternative to histological studies in the follow-up of these lesions [30,41].

#### 3.1.2. Imiquimod and Other Topical Drugs: Retinoic Acid Derivates, 5-Fluorouracil

As noted previously, to increase the inflammatory reaction, imiquimod can be combined with other topical products. Among them, the most used topical products are retinoic acid derivates. Both tazarotene (0.05% or 1%, more commonly prescribed in gel than cream) or tretinoin (0.1% cream) have been used with different schedules in some patients in the studies shown in Table 1, most commonly as pre-treatment 2–4 weeks before the start of imiquimod or concomitantly 2 days per week [29,45,49,60,62,63].

Topical retinoids enhance imiquimod penetration through the disruption of the stratum corneal barrier, which explains how they are able to induce a more potent inflammatory response [29]. One randomised controlled clinical trial by Hyde et al., which compared the efficacy of imiquimod 5% cream alone vs. imiquimod 5% plus tazarotene 1% gel 2 days per week in the treatment of 91 MIS LM-subtype, showed a significant increase in clinical inflammatory response in the group treated with tazarotene (60%, 25/42 cases vs. 81%, 30/37 cases). Moreover, the histological clearance rate achieved in the group with the combined therapy was greater, although the difference was not significant (64%, 27/42 vs. 78%, 29/37) [29].

Another topical drug that has reportedly been used in combination with imiquimod for the treatment of MIS is 5-fluorouracil (5-FU). Some studies have analysed the effectiveness of this combination in the treatment of CM metastases when surgery is contraindicated or not possible [64]. To our knowledge, only one report with 2 MIS treated with imiquimod 5% cream, 5-FU 2% solution, and tretinoin 0.1% cream has been published, achieving complete histological clearance confirmed by biopsies 3 months after therapy [62]. It is thought that the different mechanism of action of 5-FU (antimetabolite) and imiquimod (immune response booster) could lead to synergic antitumour activity [62].

**Table 1 cancers-15-04468-t001:** Studies (excluding case reports) using topical imiquimod for the treatment of melanoma in situ (MIS), including lentigo maligna (LM), as primary therapy.

Studies (Excluding Case Reports) Using Topical Imiquimod for the Treatment of Melanoma In Situ (MIS), Including Lentigo Maligna (LM), as Primary Therapy								
Study	Type of Study	Number of Lesions Analysed	Initial Regimen	Clinical Clearance	Histological Clearance	Mean Follow-Up (Months)	Recurrence	LMM
Naylor et al. (2003) [65]	Non-controlled open-label study	28	Imiquimod 5% 7 days/week 12 weeks	CR: 26 (93%) NR: 2 (7%)	Yes: 26 (93%) No: 2 (7%)	12	0	0
Fleming et al. (2004) [66]	Case series	6	Imiquimod 5% 7 days/week 6 weeks	CR: 2 (33%) PR: 3 (50%) NR: 1 (17%)	All by excisions: Yes: 4 (67%) No: 2 (33%)	N/A	0	0
Powell and Russell-Jones (2004) [67]	Case series	2	Imiquimod 5% 3 days/week 12 weeks	CR: 2 (100%) NR: 0	Yes: 2 (100%) No: 0	14	0	0
Powell et al. (2004) [68]	Non-controlled open-label study	12	Imiquimod 5% 3 days/week 6 weeks	CR: 7 (585) PR: 5 (42%)	Yes: 10 (83%) No: 2 (17%)	6	0	0
Ray et al. (2005) [69]	Case series	3	Imiquimod 5% 3–7 days/week 6–12 weeks	CR: 3 (100%) NR: 0	Yes: 3 (100%) No: 0	8	0	0
Wolf et al. (2005) [70]	Case series	6	Imiquimod 5% 7 days/week 9 weeks (mean)	CR: 6 (100%) NR: 0	Yes: 6 (100%) No: 0	10	0	0
Spenny et al. (2007) [71]	Case series	10	Imiquimod 5% 2–7 days/week 16 weeks (mean)	CR: 10 (100%)	Yes: 7 (100%) No: 0	18	0	0
Mahoney et al. (2008) [72]	Case series	7	Imiquimod 5% 5 days/week 12 weeks	CR: 6 (86%) NR: 1 (14%)	Yes: 6 (86%) No: 1 (14%)	19	0	0
Cotter et al. (2008) [49]	Case series	40	Imiquimod 5% 5 days/week 12 weeks	CR: 33 (83%) PR: 7 (17%)	All by staged excisions: Yes: 30 (75%) No: 10 (25%)	18	0	1
Buettiker et al. (2008) [48]	Non-controlled open-label study	34	Imiquimod 5% Variable: twice daily to 3 days/week 7 weeks (mean) +/− Tazarotene 0.1% gel	CR: 28 (82%) PR: 6 (18%)	Biopsied only the patients with PR: Yes: 6/6 (100%) No: 0/6	17	1	1
de Troya-Martín et al. (2008) [73]	Case series	2	Imiquimod 5% 5 days/week 12 weeks	CR: 2 (100%) NR: 0	Yes: 2 (100%) No: 0	36	0	0
Micali et al. (2008) [74]	Case series	2	Imiquimod 5% 5 days/week 16 weeks	CR: 2 (100%) NR: 0	N/A	30	0	0
Powell et al. (2009) [54]	Case series	48	Imiquimod 5% 3 days/week 6 weeks	CR: 37 (77%) PR: 2 (4%) NR: 9 (19%)	Yes: 37 (77%) No: 11 (23%)	49	0	1
Demirci et al. (2010) [75]	Case series	5	Imiquimod 5% 5–7 days/week 32 weeks (mean)	CR: 3 (60%) PR: 2 (40%)	N/A	20	0	0
van Meurs et al. (2010) [76]	Prospective cohort study	10	Imiquimod 5% 3–7 days/week 8–12 weeks (mean)	CR: 9 (90%) NR: 0	Yes: 9 (90%) No: 0	31	3	0
Missall et al. (2010) [77]	Case series	15	Imiquimod 5% 5–7 days/week Until clinical complete response (mean 12 weeks)	CR: 15 (100%) NR: 0	Yes: 15 (100%) No: 0	16	0	0
Ly et al. (2011) [52]	Non-controlled open-label study	38	Imiquimod 5% 5 days/week 12 weeks	CR: 20 (53%) NR: 18 (47%)	All by wide excisions: Yes: 20 (53%) No: 18 (47%)	N/A	N/A	N/A
Hyde et al. (2012) [29]	Randomised clinical trial	79	- 42 patients: Imiquimod 5% 5 days/week 12 weeks - 37 patients: Same + Tazarotene 0.1% gel 2 days/week 12 weeks	N/A	All by staged excisions: - Yes: 27 (64%) No: 15 (36%) - CR: 29 (78%) - No: 18 (22%)	42	0	1
Wong et al. (2012) [78]	Retrospective and prospective cohort study	27	Imiquimod 5% 3 days/week 17–20 weeks	CR: 20 (74%) NR: 7 (26%)	Yes: 20 (74%) No: 7 (26%)	41	0	0
Alarcon et al. (2014) [79]	Non-controlled open-label study	20	Imiquimod 5% 5 days/week 8 weeks	N/A	Yes: 15 (75%) No: 5 (25%)	34	0	0
Guitera et al. (2014) [59]	Case series	39	Imiquimod 5% 5 days/week 12 weeks	CR: 19/28 (68%) PR/NR: 9/28 (32%)	Biopsied only the patients with suspicious dermoscopy or RCM (n = 9): Yes: 3 No: 6	>12	3	1
Kirtschig et al. (2015) [50]	Prospective cohort study	24	Imiquimod 5% 7 days/week 14 weeks (mean)	CR: 20 (83%) NR: 4 (17%)	Yes: 24 (100%) No: 0	39	1	0
Swetter et al. (2015) [56]	Case series	22	Imiquimod 5% 3–5 days/week 12 weeks (mean)	CR: 16 (73%) NR: 2 (9%)	N/A	40	0	1
Elia et al. (2016) [24]	Case series	6	Imiquimod 5% 7 days/week 16 weeks (mean)	CR: 5 (83%) NR: 1 (17%)	Yes: 5 (83%) No: 1 (17%)	24	0	0
Gautschi et al. (2016) [58]	Non-controlled open-label study	89	Imiquimod 5% 7 days/week Once/twice daily Until inflammatory response	N/A	N/A	4.8 years	16	0
Kai et al. (2016) [61]	Case series	40	Imiquimod 5% 3 days/week 6 weeks	N/A	Yes: 27 (67%) No: 11 (28%)	7.5 years	0	0
Park et al. (2017) [60]	Case series	12	Imiquimod 5% 7 days/week 6–8 weeks or Tazarotene 0.1% cream daily 2 weeks, followed by imiquimod 5% on weekends 12 weeks	N/A	Yes: 11/11 (100%) No: 0	5.5 years	1	1
Marsden et al. (2017) [46]	Non-controlled open-label study	28	Imiquimod 5% 5 days/week 12 weeks (60 applications)	CR: 13/28 (46%) PR: 11/28 (39%) NR: 4/28 (14%)	All by excisions: Yes: 10/27 (37%) Partial regression: 10/27 (37%) No: 7/27 (26%)	N/A	N/A	N/A
Flores et al. (2018) [40]	Prospective cohort study	52	Imiquimod 5% 5 days/week 8 weeks	N/A	All by staged excisions: Yes: 43 (83%) No: 9 (17%)	N/A	N/A	N/A
Astorino et al. (2018) [80]	Case series	2	Imiquimod 5% 5 days/week Twice a day (in occlusion 12 h in the evening) Alternate week for 5 weeks	CR: 2 (100%) NR: 0	N/A	30	0	0
Tio et al. (2018) [28]	Case series	3	Imiquimod 5% 7 days/week 12 weeks	CR: 3 (100%)	Yes: 2 (100%) No: 0	32	0	0
Tio et al. (2019) [39]	Prospective cohort study	57	Imiquimod 5% 7 days/week 12 weeks	CR: 48 (84%) PR: 6 (11%) NR: 3 (5%)	Yes: 32/37 (86%) No: 5/37 (14%)	22.5	5	1
Papanikolau et al. (2019) [51]	Case series	33	Imiquimod 5% 5 days/week 6 weeks	CR: 17 (52%) NR: 16 (48%)	Histological study only in 11/16 patients with no clinical clearance: Yes: 4/11(36%) No: 7/11 (64%)	4.1 years	0	3
Brand et al. (2019) [41]	Non-controlled open-label study	34	Imiquimod 5% 6 weeks	CR: 33 (97%) NR: 1 (3%)	Yes: 33 (97%) No: 1 (3%)	N/A	7	0
Halse et al. (2020) [81]	Non-controlled open-label study	27	Imiquimod 5% 5 days/week 12 weeks	N/A	All by complete excisions: Yes: 16 (59%) No: 11(41%)	N/A	N/A	N/A
Coco et al. (2021) [82]	Case series	3	Imiquimod 5% 5 days/week 12 weeks	CR: 3 (100%) NR: 0	Yes: 2/2 (100%) No: 0	18	0	0
Lallas et al. (2021) [44]	Case series	56	Imiquimod 5% 7 days/week 6–13 weeks	CR: 40 (72%) PR: 12 (21%) NR: 4 (7%)	N/A	33.6	N/A	N/A
Chambers et al. (2021) [45]	Case series	71	Imiquimod 5% 5 days/week 12 weeks +/− Pretreatment using daily Tazarotene gel 0.1% 2 weeks	CR: 62 (87%) PR/NR: 9 (13%)	N/A	5.1 years (median) 6.2 years (mean)	7	0
Soenen et al. (2022) [30]	Randomised clinical trial	21 imiquimod 19 placebo	Imiquimod 5% 5 days/week 4 weeks	N/A	Yes: 13 (62%) No: 8 (38%)	N/A	N/A	N/A
Poveda-Montoya (2022) [53]	Case series	8	Imiquimod 5% 7 days/week 7 weeks (mean) +/− Tazarotene 0.05% gel	CR: 8 (100%) NR: 0	N/A	77	2	0
Kwak et al. (2022) [43]	Case series	24	Imiquimod 5% 4 days/week 12 weeks +/− Concurrent cryotherapy (3 cases)	After a mean of 43 months follow-up: CR: 19 (79%) NR: 5(21%)	N/A	43	5	0
Seyed Jafari et al. (2023) [42]	Prospective cohort study	114	Imiquimod 5% Twice daily (mean) 4 weeks (mean)	CR: 111 (97%) NR: 3 (3%)	N/A	96 (8 years)	23 (21%)	5

CR, complete response; N/A, not available; NR, no response; PR, partial response; RCM, reflectance confocal microscopy.

#### 3.1.3. Imiquimod and Surgery

International clinical practice guidelines for melanoma consider the use of topical imiquimod as an adjuvant therapy in selective patients with MIS LM-subtype and positive margins after surgery [17,18,19]. Adjuvant imiquimod therapy for LM after surgery in cases with histological affected or negative but narrow margins, has reported high clinical and histological clearance rates, ranging between 93 and 95% in the large series [43,45,56,83,84] (Table 2). The long-term recurrence rate was estimated between 6.5 and 7% in two recent retrospective studies with 93 and 71 LM and a mean follow-up of 2.5 and 3 years, respectively [43,44]. Another study, with a mean follow-up of 5.1 years, reported a recurrence rate of 3% developed at mean 2.9 years [56]. This is a lower recurrence rate than previously reported in LM treated with WE (6–20%) [85,86,87], and closer to that reported in those treated by surgery with microscopic margin control, both MMS and SE (0–3%) [85,86,87,88].

Kwak et at., looking for clinical or histological markers associated with a higher rate of response after imiquimod adjuvant therapy of LM, demonstrated that positive surgical margins were significantly associated with a decreased rate of clearance compared to patients with narrow margins (83.3 vs. 100%) [43]. Moreover, as occurred in the primary treatment with imiquimod, the presence of inflammatory response during treatment was associated with an increased clearance rate (95% vs. 77%) [43].

As in primary use of imiquimod for MIS, it has been shown that the best regimen for adjuvant imiquimod consists of 5–7 applications per week for 12 or more weeks (at least 60 applications), with a 2 cm margin of clinically normal skin [19,38,44,45,56,83]. Regardless of the protocol used, it is important to adjust the application regimen to achieve an inflammatory reaction and, if necessary, tazarotene or tretinoin may be added [53,83].

Imiquimod has also been used as a neoadjuvant therapy in LM, before a complete excision. SE and MMS of LM often requires multiple stages and can result in significant cosmetic and/or functional morbidity. Different studies have demonstrated that this off-label use of imiquimod is useful for decreasing the final surgical defect size required to confirm negative histological margins [29,89,90]. A recent study, with 334 patients with LM treated with imiquimod 5% 5 days per week during 8–12 weeks before staged excision, reached a median final margin of 2 mm with a recurrence rate of 3.9% with a mean recurrence time of 4.3 years (mean follow-up of 5.5 years) [90]. The authors concluded that neoadjuvant imiquimod prior to conservatively applied SE allowed a rate of recurrence similar to reported recurrence rates with standard SE. Sampson et al. proposed a strategy to reduce the necessary margin for complete clearance that consists of two steps [89]. Firstly, they removed the LM with excisional biopsy and closed it with a purse-string suture, which ruled out an invasive component. Subsequently, neoadjuvant imiquimod 5% 5 days per week for 8 weeks was applied, followed by a staged excision with 2 mm margins. They described that this technique allowed for a reduction in the required margins of 71% compared with average published margins for LM. Hyde et al. recommended a similar protocol, extending the imiquimod pre-treatment to 12 weeks [29]. 

In addition, Hyde et al. pointed out that excised LM margins treated previously with imiquimod 5% are much easier to interpret for a pathologist because atypical melanocytic hyperplasia, characteristic of skin with chronic sun exposure, is greatly eliminated and makes the assessment of margins less ambiguous [29]. Recently, in a cohort of 52 patients, Flores et al. demonstrated that the neoadjuvant use of imiquimod 5% (applied 5 times weekly for 8 weeks) was statistically associated with decreased melanocyte density counts (MDC) in LM treatment sites compared with the MDC of negative control sites [40]. The decreased melanocytic hyperplasia in imiquimod-treated sites helps the pathologist in the interpretation of surgical margins in a future excision.

Long-term follow-up in patients treated with adjuvant imiquimod will be as important as in cases primarily treated with imiquimod. RCM also plays a relevant role in the assessment of recurrences, but may also be used to identify subclinical extension and delineate surgical margins’ previous excision [86,91].

**Table 2 cancers-15-04468-t002:** Studies (excluding case reports) using topical imiquimod for the treatment of melanoma in situ (MIS), including lentigo maligna (LM), as adjuvant therapy.

Studies (Excluding Case Reports) Using Topical Imiquimod for the Treatment of Melanoma In Situ (MIS), Including Lentigo Maligna (LM), as Adjuvant Therapy								
Study	Type of Study	Number of Lesions Analysed	Initial Regimen	Clinical Clearance	Histological Clearance	Mean Follow-Up (Months)	Recurrence	LMM
Kupfer-Bessaguet (2004) [92]	Case series	2 (positive margins)	Imiquimod 5% 3 days/week 12 weeks	CR: 2 (100%) PR: 0	Yes: 2 (100%) No: 0	14	0	0
Spenny et al. (2007) [71]	Case series	2 (positive margins)	Imiquimod 5% 2–7 days/week 16 weeks (mean)	CR: 2 (100%)	Yes: 1 (100%)	18.5	0	0
Swetter et al. (2015) [56]	Case series	36 (25 positive margins; 11 narrow-margin resection)	Imiquimod 5% 3–5 days/week 12 weeks (mean)	After a mean of 43 months of follow-up: CR: 34 (94%) NR: 2 (6%)	N/A	43	2	0
Pandit et al. (2015) [83]	Case series	21 (positive margins)	Imiquimod 5% 5 days/week 12 weeks	N/A	Yes: 20 (95%) No: 1 (5%)	24	0	0
Elia et al. (2016) [24]	Case series	2 (positive margins)	Imiquimod 5% 7 days/week 16 weeks (mean)	CR: 2 (100%) NR: 0	Yes: 2 (100%) No: 0	15	0	0
Tsay et al. (2019) [84]	Non-controlled open-label study	16 (positive margins)	Imiquimod 5% 3–5 days/week 6 weeks	N/A	All by narrow re-excisions: Yes: 14 (93%) No: 1 (7%)	14.3	0	0
Lallas et al. (2021) [44]	Case series	93 (Group 1: 71 narrow/wider margins and negative margins; Group 2: 22 positive margins)	Imiquimod 5% 7 days/week 6–13 weeks	N/A	N/A	33.3 Group 1: 32.5 Group 2: 34	6 (6.5%) Group 1: 4 (5.6%) Group 2: 2 (9.1%)	0
Chambers et al. (2021) [45]	Case series	32 (narrow margins or positive margins)	Imiquimod 5% 5 days/week 12 weeks +/− Pretreatment using daily Tazarotene gel 0.1% 2 weeks	CR: 30 (94%) PR/NR: 2 (6%)	N/A	5.1 years (median) 6.2 years (mean)	1	0
Poveda-Montoya (2022) [53]	Case series	4 (narrow margins)	Imiquimod 5% 7 days/week 7 weeks (mean) +/− Tazarotene 0.05% gel	CR: 4 (100%) NR: 0	N/A	55	0	0
Kwak et al. (2022) [43]	Case series	42 LM + 29 LMM (25 positive margins; 42 narrow margins)	Imiquimod 5% 4 days/week 12 weeks +/− Concurrent cryotherapy (1 case)	After a mean of 37 months of follow-up: CR: 66 (93%) NR: 5 (7%)	N/A	37	5	1

CR, complete response; N/A, not available; NR, no response; PR, partial response.

#### 3.1.4. Imiquimod and Cryosurgery (Immunocryosurgery)

Immunocryosurgery refers to the combination of imiquimod and cryosurgery (usually with liquid nitrogen). It has been described as a non-surgical alternative therapy for MIS LM-subtype [93]. This combination increases the inflammatory response and thus the likelihood of a complete response [94]. Different case reports and case series showed good clearance rates and functional/cosmetic results [24,43,48,95,96,97,98].

Many treatment regimens have been described, using cryosurgery before, during, or after imiquimod therapy, with different results. Matas-Nadal et al. and Bassukas et al. used imiquimod 5% cream once daily for 3 weeks, followed by one session of cryosurgery with 2 freeze-thaw cycles on the LM plus 1 cm margin, and then complete treatment with another 6 months of imiquimod, applied 3 days per week [96,98]. Bratton et al. proposed a different regimen, using imiquimod 5% 3 times weekly for 3 months and an additional 5 times weekly after 3 months, for a total course of 4 months of therapy, combined with 3 sessions of cryotherapy at 1, 2, and 3 months [97].

Liquid nitrogen reaches very low temperatures of up to −196 °C, which can cause significant local adverse effects [95]. In order to reduce local reactions in sensitive areas, Oro-Ayude et al. described a different technique to treat two periocular LM using an ophthalmic cryosurgery system [95]. This system uses nitrous oxide, which has a freezing temperature of −89 °C, not as intense as that of liquid nitrogen, resulting in lower patient morbidity [95]. These authors proposed performing a unique session of cryosurgery using 3 freeze–thaw cycles on the lesion plus 1 cm of clinically normal skin, followed by 8 weeks of imiquimod 5% application 5 days per week. After 1.5 years of follow-up, no relapses were detected.

#### 3.1.5. Imiquimod and Laser Therapy

De Vries and Greveling et al. proposed a novel treatment combination for LM consisting of ablative laser therapy (CO2 laser or erbium-doped yttrium aluminium garnet laser) with a 2–3 cm margin of adjacent skin, followed by 6 weeks of topical imiquimod 5% application 5 days per week [99,100]. Vries et al. reported the absence of recurrence after a mean of 22 months of follow-up in a cohort of 12 patients [99]. Subsequently, Greveling et al. expanded this cohort by 23 patients treated with the same scheme, bringing the size of the case series to 35 LM, with a mean follow-up of 19 months. The cumulative incidence of recurrence after 1 year was 9.4%, and after 3 years it was 23.5% [100]. 

Laser therapy would eliminate the majority of atypical melanocytes by removing the epidermis and papillary dermis, but it would also allow deeper penetration of imiquimod, bringing it closer to its target cell (dendritic cell) to destroy residual atypical melanocytes [100]. This would theoretically lead to increased inflammation, associated with a higher response rate.

In a systematic review of non-surgical therapies for LM, Read et al. described mean recurrence rates of 34.4% for laser therapy in monotherapy [57]. Apparently, the combination therapy with imiquimod reduces the risk of recurrence [100]. 

### 3.2. Intralesional Interferon-Alfa (IL-INF-α)

The effect of IL-INF-α, another type of local immunotherapy that has been used for the successful treatment of MIS in different case reports, is twofold. Firstly, it acts as an immunomodulator, increasing the expression of histocompatibility complex class-1 and thus the cytotoxic activity of natural killer cells, while also enhancing antigen expression on the tumour cell surface. In addition, it has an antiproliferative action on melanoma cells [101]. Associated side effects include a local reaction with erythema and skin peeling, and systemic symptoms as fever, chills, or flu-like syndrome, which are frequent but transitory [101].

Cornejo et al. achieved a complete clinical and/or histological clearance in 100% of 11 LM treated with IL-INF-α. The protocol used included 3 million units for tumours ≤ 2.5 cm and 6 million units for tumours > 2.5 cm, 3 times per week with a mean number of injections of 19 (range 12–29). No recurrence registered after a mean follow-up of 15.4 months [101]. Carucci et al. reported a case of recurrent MIS on the eyelid, treated with the same dosage of IL-INF-α 3 times per week, for a total of 39 million units, with histological clearance observed in four post-treatment biopsies [102]. Turner et al. published the case of a woman with xeroderma pigmentosum and 10 MIS, with 5 treated with IL-INF-α 3 times per week for 3 weeks and 5 with a placebo [103]. All the lesions were excised by MMS and 100% of those treated with IL-INF-α revealed no evidence of MIS, despite using a lower dose than the aforementioned studies (1 vs. 3 million units).

### 3.3. Ingenol Mebutate (IM)

IM is a diterpene ester derivative from the plant *Euphorbia peplus* which has been approved as a topical agent in the treatment of actinic keratosis [22]. It is applied on the lesion once daily over 3 consecutive days to be effective and its side effects are related to local inflammation. Although its mechanism of action is still not well known, previous studies have shown that it is able to induce apoptosis and increase the production of pro-inflammatory cytokines and antitumoural cytotoxic T cells (immunomodulatory effect) [22,104,105]. This supports why this drug has been tested on different superficial skin cancers such as MIS.

IM has not been demonstrated to be effective on MIS lesions. A non-controlled open-label study of 12 LM treated with IM 0.015% gel for 3 days achieved disappointing results [106]. Only 2 patients presented clinical and histological response at 2 months, but 1 of them relapsed at 8 months. Moreover, no correlation between IM-induced inflammation and clinical or histological clearance was observed. Similar results were published in a case report of a LM also treated with IM 0.015% gel for 3 days; although there was apparent inflammation, no clinical and histological clearance was observed [107]. To our knowledge, only one case report of recurrent MIS treated with the same schedule as the aforementioned studies and 6 months of follow-up achieved complete clinical and histological clearance [108].

Larger studies are needed to evaluate the effectiveness of IM in MIS lesions and in melanocytic tumours in general. However, the production and marketing of this drug was halted in 2020 due to a possible increased risk of squamous cell carcinoma in patients with actinic keratosis treated with IM [109,110].

## 4. Strengths and Limitations

This was an extensive review of the published literature on the use of topical and intralesional immunotherapy in the treatment of MIS, with theoretical and practical purposes. Not only the use of immunotherapy in monotherapy was analysed, but also the different combinations with other therapeutic alternatives, both in adjuvant and neoadjuvant therapy. 

It is also important to highlight the methodological limitations of this review study as it is not a systematic review. Likewise, we found it difficult to make comparisons in therapeutic results between the different studies, since there was no uniformity in terms of previous treatments, the treatment regimen under study, the duration of therapy, and the assessment of outcomes.

## 5. Conclusions and Future Directions

In conclusion, local immunotherapy is frequently used in clinical practice, and the success rate in real-life experience, especially using imiquimod, is encouraging. Surgical excision remains the gold standard in the treatment of MIS. Nevertheless, the response and recurrence rates reported in MIS treated with imiquimod as monotherapy, or as an adjuvant after surgery with affected or narrow margins, make imiquimod a reliable therapeutic alternative in selected cases. Adjuvant and neoadjuvant use of imiquimod seems to improve the recurrence rate compared to conventional WE in LM. This drug also allows for combined treatments with other therapeutic alternatives (topical retinoids, cryosurgery, laser therapy) in order to increase its effectiveness.

In the future, we may be able to predict which patients will respond and have a low risk of recurrence with imiquimod treatment, so that physicians can make an appropriate patient selection. Gautschi et al. proposed assessing the number of melanocytes in the diagnostic biopsy to predict the response to imiquimod treatment, due to the association found between the risk of relapse and number of melanocytes per millimetre in the original biopsy [58]. Recently, Halse et al. identified a gene signature in the initial diagnostic biopsy of LM that seems to be unique to imiquimod responders [81]. These findings need to be corroborated with other studies but may be a potential biomarker of response.

Regarding the use of IL-INF-α and IM for the treatment of MIS, there is not as much scientific evidence as for imiquimod. Studies with a larger sample size or a head-to-head clinical trial compared with, or as an adjunct to, surgical excision are needed to provide data relating to the actual response and recurrence rate of MIS treated with the aforementioned types of immunotherapy.

## Data Availability

The data presented in this study are available in this article.

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
