# Peer review of "Topical and Intralesional Immunotherapy for Melanoma In Situ: A Review"

_cancers, 2023, doi:10.3390/cancers15184468_

Round 1

Reviewer 1 Report

The authors performed a review of the current literature with the aim of investigating the applications of immunotherapy in MIS, either in monotherapy or in combination with other therapeutic alternatives. The manuscript is interesting and well-written. I have only few comments:

- Introduction: nothing to add

- Material and Methods section: this section should be added in order to explain how you performed literature review

- Strengths and limitations: this section should be added

Reviewer 2 Report

This is an excellent review.

One comment:  A description of the types of patients where surgery might not be feasible medically is appropriate (ie the patient with End stage renal disease, Heart Failure etc).  Some of these surgeries can be complex and a lot of people incorrectly think that excision of a superficial lesion is always possible.   
